# Trade-offs in Ensembling, Merging and Routing Among Parameter-Efficient Experts

**Sanae Lotfi**                                                              *sl8160@nyu.edu*
*New York University*

**Lucas Caccia**                                                   *lpagecaccia@microsoft.com*
*Microsoft Research*

**Alessandro Sordoni**                                                *alsordon@microsoft.com*
*Microsoft Research*

**Jordan T. Ash**                                                  *ash.jordan@microsoft.com*
*Microsoft Research*

**Miroslav Dudik**                                                     *mdudik@microsoft.com*
*Microsoft Research*

**Reviewed on OpenReview:** *https://openreview.net/forum?id=bnRCvRtZv5*

## Abstract

While large language models (LLMs) fine-tuned with lightweight adapters achieve strong performance across diverse tasks, their performance on individual tasks depends on the fine-tuning strategy. Fusing independently trained models with different strengths has shown promise for multi-task learning through three main strategies: ensembling, which combines outputs from independent models; merging, which fuses model weights via parameter averaging; and routing, which integrates models in an input-dependent fashion. However, many design decisions in these approaches remain understudied, and the relative benefits of more sophisticated ensembling, merging and routing techniques are not fully understood. We empirically evaluate their trade-offs, addressing two key questions: What are the advantages of going beyond uniform ensembling or merging? And does the flexibility of routing justify its complexity? Our findings indicate that non-uniform ensembling and merging improve performance, but routing offers even greater gains. Surprisingly, uniform ensembling with no learning outperforms all merging methods, suggesting that mode connectivity constraints fundamentally limit parameter-space fusion in the multi-task setting. Meanwhile, SGD-optimized routing achieves the best non-oracle performance with less than 1% inference compute overhead. To mitigate the computational cost of routing, we analyze expert selection techniques, showing that clustering and greedy subset selection can maintain reasonable performance with minimal overhead. These insights advance our understanding of model fusion for multi-task learning.

## 1 Introduction

The growing availability of publicly fine-tuned large language models (LLMs) presents an opportunity to integrate existing models for multi-task learning without requiring explicit knowledge of their original training data. Platforms like the HuggingFace model hub[1] now host over a million publicly available models, many of which have been finetuned via parameter-efficient methods such as LoRA (Hu et al., 2021) on diverse tasks, starting from the same pretrained model. LoRA significantly reduces the number of trainable parameters

---

[1] https://huggingface.co/models

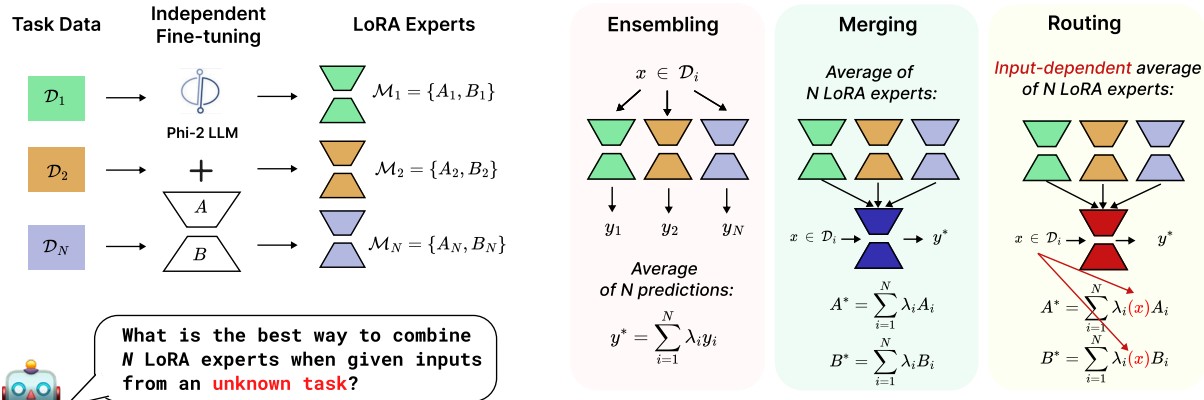

(a) Library of LoRA experts        (b) Comparison of ensembling, merging, and routing

Figure 1: **Model fusion of parameter-efficient experts for task-agnostic multi-task learning. (a)**: We use a publicly available library of LoRA experts, each fine-tuned independently on Flan v2 tasks from the same Phi-2 pretrained LLM (Ostapenko et al., 2024). These experts provide a diverse foundation for multi-task learning. **(b)**: Comparison of three model fusion approaches and their trade-offs. *Ensembling* aggregates outputs from the independent experts at inference. *Merging* fuses expert weights into a single model via parameter averaging. *Routing* extends merging by making the weight combination input-dependent, adapting to each input dynamically.

while largely maintaining the expressivity of fine-tuning. This raises a fundamental question: *How can we optimally integrate these experts to achieve strong, task-agnostic performance across multiple tasks while minimizing computational costs?*

One promising approach is to exploit mode connectivity, a hypothesis from loss surface literature suggesting that models fine-tuned from the same pretrained initialization remain close in parameter space (Draxler et al., 2018; Garipov et al., 2018; Frankle et al., 2020; Neyshabur et al., 2020; Benton et al., 2021; Ainsworth et al., 2022). This assumption has motivated a body of work on model merging (Matena & Raffel, 2022; Jin et al., 2022; Tam et al., 2023), where the weights of independently trained models are combined directly in parameter space to produce a new model that inherits their capabilities. Formally, given $N$ models with parameters $w_1, \ldots, w_N$, merging produces a single model $w^* = \sum_{i=1}^{N} \lambda_i w_i$ such that $\sum_{i=1}^{N} \lambda = 1$ and $\lambda_i \geq 0$. Here, we refer to input-independent merging, where the merging weights $\lambda_i$ remain fixed for all inputs, simply as merging.

An alternative approach to merging is routing, an input-dependent form of model fusion in parameter space that relaxes the mode connectivity assumption by allowing for the coefficients $\lambda_i$ to depend on the input. Routing is well-known in Mixture-of-Experts (MoE) (Shazeer et al., 2017), but recently has been also applied to "MoErge" separate expert models, trained independently (Yadav et al., 2024a; Ostapenko et al., 2024; Muqeeth et al., 2024; Tam et al., 2024c). This approach can selectively exclude certain experts or adjust their corresponding contribution depending on the input. In the extreme case where only one expert is assigned a nonzero weight, routing reduces to standard expert selection.

While mode connectivity has been validated in single-task settings, work on multi-task learning for vision suggests that this assumption may not hold when models are fine-tuned on diverse tasks (Yamada et al., 2023). If task-specific fine-tuning moves models further apart in parameter space, simple merging may not always be optimal. One global model integration approach that does not require mode connectivity is ensembling, which aggregates the outputs of multiple models to improve robustness and performance. Figure 1(b) shows the difference between routing, merging and ensembling. Ensembling has been widely successful in uncertainty estimation and multi-expert modeling (Lakshminarayanan et al., 2017), but it is significantly more expensive than merging and routing because it requires $N$ forward passes at inference time.

In this work, we conduct a thorough analysis to answer the following questions that were left open by previous works (Ostapenko et al., 2024; Muqeeth et al., 2024): Does ensembling provide advantages over merging and routing that justify its computational cost? What are the benefits of moving beyond uniform ensembling or merging? Can clustering and expert refactoring reduce the number of experts while maintaining strong performance, thereby changing the trade-offs between these approaches?

To answer these questions, we compare ensembling, merging and routing for multi-task learning using a library of parameter-efficient, publicly-available LoRA experts from Ostapenko et al. (2024), fine-tuned from the Phi-2 language model (Javaheripi et al., 2023) on Flan v2 tasks (Longpre et al., 2023) as shown in Figure 1(a). We summarize our contributions as follows:

- **Careful evaluation of model integration strategies.** We compare ensembling, merging, and routing in a task-agnostic, multi-task learning setting, where task identifiers are unknown at inference. We investigate both uniform and non-uniform ensembling and merging, showing that learned weight combinations significantly improve over naive uniform approaches. We provide a quantitative cost analysis (Table 1) showing that routing adds less than 1% inference overhead while ensembling occupies the opposite end of the cost spectrum.

- **Mode connectivity and routing effectiveness.** We analyze pairwise mode connectivity, showing that while models fine-tuned from the same initialization can be linearly interpolated with low loss, routing between these models consistently outperforms merging on both individual and combined datasets. We expand this analysis to a diverse set of task pairs and report aggregate statistics, revealing that the merging-routing gap is rooted in multi-task loss surface geometry.

- **Trade-offs between routing complexity and expert selection.** We assess whether the additional flexibility of routing justifies its computational cost, finding that while routing consistently improves performance, carefully selected expert subsets can approximate its benefits at lower cost. We explore expert clustering and hierarchical merging, reducing the number of experts from 256 to 10 while maintaining strong generalization. This 96% reduction in experts also reduces the router parameter count: routing over 256 private experts would require ∼42 million router parameters, whereas our 10 experts require only ∼1.6 million parameters.

Our analysis provides a deeper understanding of the trade-offs between ensembling, merging, and routing for multi-task learning. Several of our findings are non-trivial. First, uniform ensembling (with no learning) outperforms all merging methods, including SGD-optimized variants. This is surprising given that learned merging coefficients should in principle improve over a simple uniform average in output space, but mode connectivity constraints prevent this. Second, SGD-optimized routing over the 10 MBC experts achieves the best non-oracle performance with less than 1% inference compute overhead and only ∼1.6 million router parameters (Table 1). Third, expert libraries exhibit massive redundancy: routing directly over all 256 private experts would require ∼42 million router parameters, but reducing to ∼10 experts preserves most of the performance.

## 2 Preliminaries

We consider the multi-task learning setting, where $N$ models with parameters $w_1, \ldots, w_N$ are fine-tuned independently from the same pretrained model on a set of $N$ different tasks $\mathcal{T}_1, \ldots, \mathcal{T}_N$. We particularly focus on models fine-tuned using Low-Rank Adaptation (LoRA) (Hu et al., 2021), a parameter-efficient fine-tuning approach. LoRA introduces low-rank updates to the pretrained model's weight matrices, represented as the product of two much smaller matrices $A$ and $B$, such that the update $\Delta W$ is given by $AB$, where $A \in \mathbb{R}^{n \times r}$ and $B \in \mathbb{R}^{r \times m}$, with $r \ll \min\{n, m\}$. This approach significantly reduces the number of trainable parameters by freezing the original model weights and optimizing only the low-rank matrices. As a result, LoRA achieves competitive performance on a wide range of tasks while maintaining computational and storage efficiency, making it particularly suitable for scenarios involving multiple independently fine-tuned models.

We refer to the resulting fine-tuned models as *parameter-efficient experts*, as each serves as a lightweight adapter specialized for its respective task. The goal of our work is to start with these $N$ experts and achieve the best average performance across tasks $\mathcal{T}_1$ to $\mathcal{T}_N$ through ensembling, merging, and routing approaches. Notably, we operate under the assumption that the task identifier for each input is not known during evaluation, making it necessary for our methods to generalize without explicit access to that information.

## 2.1 Leveraging a Library of LoRA Experts

We build on the library of parameter-efficient experts introduced by Ostapenko et al. (2024), which consists of 256 LoRA fine-tuned experts trained on tasks from the Flan v2 dataset (Longpre et al., 2023) using the pretrained Phi-2 model (Javaheripi et al., 2023), a 2.8 billion parameter large language model (LLM). This library, made publicly available on HuggingFace, is designed to cover a diverse range of tasks.

To improve multi-task learning performance while reducing the number of experts to a reusable subset, Ostapenko et al. (2024) introduce Model-Based Clustering (MBC). MBC identifies task similarities by computing pairwise cosine similarities between LoRA parameter vectors, grouping tasks into clusters based on the resulting similarity matrix, and training a single expert on each cluster. This approach consolidates the original 256 experts into 10 cluster-based experts, which are publicly available on HuggingFace. It therefore strikes a trade-off between the computational cost of storing and combining these experts and the performance improvements gained through model integration. We refer to the individual fine-tuned models as *private experts* and the fine-tuned models for each cluster as *MBC experts*. For computational feasibility, we use the 10 MBC experts as the basis for comparing ensembling, merging, and routing approaches in Section 3 and Section 4, while exploring alternative expert selection strategies in Section 5.

## 2.2 Model Fusion Approaches

As discussed in the previous section, we consider the setting where we have access to $N$ LoRA experts, each fine-tuned independently on $N$ distinct tasks. Since all experts share the same pretrained model weights, they can be fully represented by the low-rank adaptation matrices used during fine-tuning. Formally, the parameters of each expert $\mathcal{M}_i$ at a given layer $l$ can be written as $(A_i^{(l)}, B_i^{(l)})$, where $A_i^{(l)} \in \mathbb{R}^{m \times r}$ and $B_i^{(l)} \in \mathbb{R}^{r \times n}$ are the LoRA matrices injected into the frozen pretrained model. For clarity, we omit the layer index $l$ in the following sections and refer to each expert using its LoRA parameters $(A_i, B_i)$.

We now define the model fusion approaches considered in this work.

**Ensembling.** Given an input sequence $x_{<t}$, each expert $\mathcal{M}_i$ produces a conditional probability distribution over the next token $p(x_t \mid x_{<t}; A_i, B_i)$. The final ensembling prediction is computed as a weighted sum of these probabilities $p(x_t \mid x_{<t}) = \sum_{i=1}^{N} \lambda_i p(x_t \mid x_{<t}; A_i, B_i)$ where the ensembling coefficients satisfy $\sum_{i=1}^{N} \lambda_i = 1$ and $\lambda_i \geq 0$.

**Merging.** Merging fuses multiple experts into a *single model* by averaging their LoRA parameters $A^* = \sum_{i=1}^{N} \lambda_i A_i$, $B^* = \sum_{i=1}^{N} \lambda_i B_i$, where $\sum_{i=1}^{N} \lambda_i = 1$ and $\lambda_i \geq 0$. While merging, by definition, uses input-independent coefficients, $\lambda_i$ does not have to be similar across all experts and layers. In later sections, we explore layer-dependent merging, where the merging coefficients $\lambda_i$ vary per layer, compared to global merging, where a single set of $\lambda_i$ is applied across all layers.

**Routing.** Routing generalizes merging by making the fusion coefficients *input-dependent*, such that $A^*(x_{<t}) = \sum_{i=1}^{N} \lambda_i(x_{<t}) A_i$ and $B^*(x_{<t}) = \sum_{i=1}^{N} \lambda_i(x_{<t}) B_i$. Routing allows the final model to selectively combine experts based on the input.

Table 1 summarizes the computational trade-offs for all methods studied in this paper. We derive these numbers for the Phi-2 setup ($d_{\text{model}} = 2560$, $L = 32$ layers, LoRA rank $r = 4$). In the main experiments (Sections 3–4), we use $N = 10$ MBC experts; HC and Arrow HC in Section 5 cluster the 256 private experts into 10 groups. Phi-2 uses a fused QKV projection (Wqkv) and a separate output projection (out_proj), giving 2 LoRA-modified layers per transformer block. The base model cost per token per layer is approximately 78.6 million FLOPs. Full derivations are provided in Appendix A.1.

Table 1: **Computational trade-offs for all model fusion methods.** All costs are measured in linear-projection FLOPs per token per layer (omitting attention-core compute, LayerNorm, and residuals, which are constant across methods). The main methods (ensembling, merging, SGD routing) use $N = 10$ MBC experts; HC and Arrow HC cluster the 256 private experts into 10 groups. $C_{\text{base}} \approx 78.6$ million FLOPs per token per layer. $P_{\text{LoRA}} \approx 61$ thousand FLOPs per layer ($\sim 0.08\%$ of $C_{\text{base}}$). Training cost is reported per step; all SGD methods train for 5 epochs (see Appendix A). See Appendix A.1 for the full derivation.

| Method | Fwd. Passes | Inference Overhead | Extra Learned Params | Training Cost (per step) |
|---|---|---|---|---|
| Uniform Ensembling | $N$=10 | $N \times C_{\text{base}}$ (10×) | 0 | 0 (zero-shot) |
| SGD Ensembling | $N$=10 | $N \times C_{\text{base}}$ (10×) | 10 scalars | $N \times C_{\text{base}}$ |
| Distillation | 1 | $P_{\text{LoRA}}$ ($\sim 0.08\%$) | $\sim 2$ million | $N \times C_{\text{base}} + 3 \times C_{\text{base}}$ |
| Uniform Merging | 1 | $P_{\text{LoRA}}$ ($\sim 0.08\%$) | 0 | 0 (zero-shot) |
| SGD Merging (global) | 1 | $P_{\text{LoRA}}$ ($\sim 0.08\%$) | 10 scalars | $3 \times C_{\text{base}}$ |
| SGD Merging (per-layer) | 1 | $P_{\text{LoRA}}$ ($\sim 0.08\%$) | 320 scalars | $3 \times C_{\text{base}}$ |
| SGD Routing ($N$=10) | 1 | $\sim 0.85\%$ | $\sim 1.6$ million | $3 \times (C_{\text{base}}+0.85\%)$ |
| HC (256→10) | 1 | $\sim 0.85\%$ | $\sim 1.6$ million + merge coefs. | $3 \times (C_{\text{base}}+0.85\%)$ |
| Arrow HC (256→10) | 1 | $\sim 0.85\%$ | $\sim 1.6$ million (Arrow) + merge coefs. | $3 \times (C_{\text{base}}+0.85\%)$ |

## 2.3 Evaluation

We evaluate the different ensembling, merging, and routing approaches by computing the loss on the test set of all 256 tasks, even when using only the 10 MBC experts. This ensures a comprehensive assessment of each method's ability to generalize across the full task distribution. To account for variability in performance, we also compute and report the standard error, providing a measure of statistical reliability and robustness.

## 2.4 Baselines

Before evaluating our ensembling, merging, and routing techniques, we first establish baseline approaches that serve as reference points for comparison.

**Oracle Baseline.** The oracle baseline assumes perfect knowledge of the task identifier, which allows for exact routing to the appropriate expert. For private experts, this means selecting the model that was fine-tuned on the task corresponding to the given input. For MBC experts, we route to the expert associated with the cluster containing the task to which the input belongs. Since our primary setup assumes no access to task identifiers, this serves as an upper-bound reference for performance.

**Shared Expert Baseline.** As an additional baseline, we consider a single shared LoRA expert fine-tuned on all 256 Flan v2 tasks. This baseline provides a direct comparison to model fusion approaches, testing whether a unified multi-task model can match or exceed the performance of specialized experts. If ensembling, merging and routing significantly outperform this baseline, this suggests that fusing task-specific experts is more effective than training a single model on all tasks.

**Arrow Baseline.** We include Arrow as a baseline for merging and routing since it is the strongest performing method for these experts and for the Flan v2 dataset in Ostapenko et al. (2024). Arrow is a zero-shot routing mechanism that dynamically selects experts without requiring joint training or explicit task labels. Specifically, for each LoRA expert $(A_i, B_i)$, Arrow estimates the routing matrix $W_\ell$ for each layer $\ell$ by computing the Singular Value Decomposition (SVD) of the outer product of $A_i$ and $B_i$: $A_i B_i^T = U_i D_i V_i^T$. The representation for expert $i$ is the first right singular vector, $V_i[:, 0]$, which is then used to initialize the routing matrix such that $W_\ell[i] = V_i[:, 0]$. At inference, for each token at layer $\ell$, given its hidden state $h_\ell$, the routing coefficients are computed as: $\lambda_i^\ell = \text{softmax}(|W_\ell[i]^\top h_\ell|)$. This approach enables modular and adaptive expert selection while maintaining computational efficiency.

# 3 Ensembling Language Models for Multi-Task Learning

A central question in model fusion is whether the improvements we could potentially get from ensembling can justify its high computational cost, which scales linearly with the number of models. In this section,

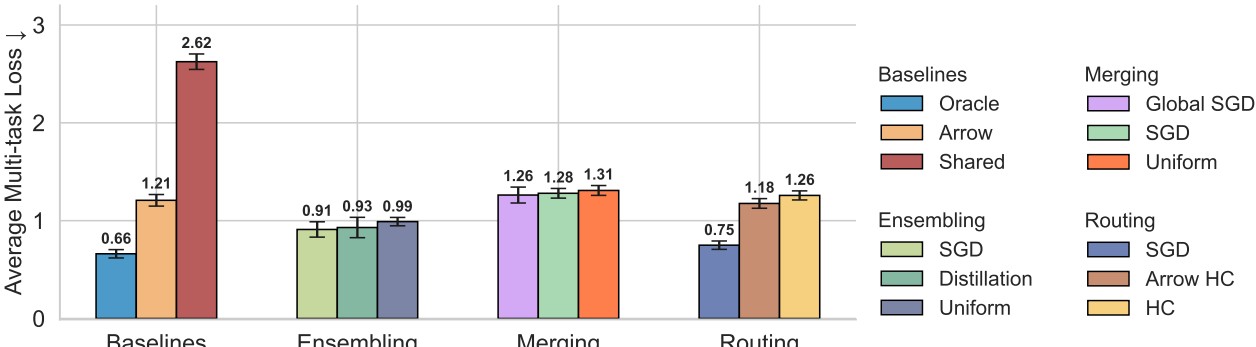

Figure 2: **Performance of different model fusion approaches for multi-task learning.** We evaluate the average multi-task test loss across 256 Flan v2 tasks for various ensembling, merging, and routing strategies, reporting the standard error across all tasks. Ensembling methods include uniform output averaging, as well as learned ensembling coefficients through stochastic gradient descent (SGD) optimization and knowledge distillation into a single model. Merging strategies involve parameter-space fusion via uniform averaging and SGD optimization of the fusion coefficients, where the learned coefficients can be either globally shared across all layers (Global SGD) or layer-specific. Routing strategies include layer-dependent routing optimization via SGD, hierarchical clustering (HC), and an optimized version of HC initialized with Arrow weights (Arrow HC). Our results indicate that ensembling outperforms merging, which may reflect limitations of the mode connectivity assumption. Routing, however, delivers the best performance among non-oracle methods. Notably, end-to-end SGD optimization consistently yields the best results across all model fusion approaches.

we evaluate the performance of uniform and learned ensembling approaches, and how they compare to the baselines introduced in Section 2.

Since we operate under the assumption that task identifiers are not available at test time, we examine how ensembling can best use multiple experts without explicit task identification. While ensembling has been widely used for single-task learning (Lakshminarayanan et al., 2017; Wang et al., 2023; Li et al., 2024), its effectiveness in multi-task learning remains underexplored.

Formally, given $N$ experts with LoRA parameters $\{(A_1, B_1), \ldots, (A_N, B_N)\}$, the ensemble prediction is given by: $p(x_t \mid x_{<t}) = \sum_{i=1}^{N} \lambda_i p(x_t \mid x_{<t}; A_i, B_i)$, where $\sum_{i=1}^{N} \lambda_i = 1$ and $\lambda_i \geq 0$. *Uniform ensembling* refers to the setting where $\lambda_i = \frac{1}{N}$ for all $i$, and *learned ensembling* to the setting where the coefficients $\lambda_i$ are optimized to minimize the average multi-task loss. Throughout this section, we use the 10 MBC experts trained on clustered Flan v2 tasks while evaluating our approaches on all 256 tasks.

In this section, we address two key questions: Can we improve accuracy by moving beyond uniform weighting in ensembling? And can we achieve a similar performance to the best ensembling approach without the computational overhead using knowledge distillation?

### 3.1 Uniform Ensembling

We begin by evaluating uniform ensembling, where all experts are assigned equal coefficients, to assess its effectiveness in the multi-task learning setting. As shown in Figure 2, uniform ensembling proves to be a strong approach, achieving competitive multi-task test loss. Notably, it outperforms all baselines except the oracle, which has access to the task ID: an advantage not available in our task-agnostic setting.

One may also ask whether ensembling logits instead of probabilities leads to better results. Additional experiments in Appendix B.4 confirm that ensembling at the probability level consistently outperforms ensembling at the logit level.

Despite its strong performance, uniform ensembling comes with a high computational cost, requiring $N$ forward passes. This raises a natural question: Can adjusting and sparsifying ensembling coefficients, $\lambda_i$, further improve multi-task test loss while maintaining computational efficiency? We explore these questions in Section 3.2 and Appendix B.6.

## 3.2 Learned Ensembling Approaches

Beyond uniform ensembling, we explore learned ensembling methods that adjust expert contributions to optimize the average multi-task performance. We present below gradient-based approaches and discuss a gradient-free alternative in Appendix B.6.

### 3.2.1 Direct optimization via SGD

We explore whether a gradient-based approach can improve ensembling by learning coefficients $\lambda$ shared across all inputs via SGD. Specifically, we optimize $\lambda$ to minimize the empirical risk over a dataset $\mathcal{D}$, which consists of training data drawn from all 256 tasks,

$$\min_{\lambda} \frac{1}{|\mathcal{D}|} \sum_{(x_{<t}, x_t^*) \in \mathcal{D}} \mathcal{L}\left(\sum_{i=1}^{N} \lambda_i p(x_t \mid x_{<t}; A_i, B_i), x_t^*\right),$$

where $\mathcal{L}$ is the cross-entropy loss function. Unlike routing, $\lambda$ remains input-independent; it is learned once and applied across all inputs.

To optimize $\lambda$, we train on data sampled from all tasks without requiring task ID access, ensuring a task-agnostic approach. At each iteration, we compute $N$ forward passes, weigh predictions according to $\lambda$, and update it via SGD to minimize multi-task test loss. As shown in Figure 2, this SGD-optimized ensembling approach improves over uniform ensembling, further closing the gap with the oracle baseline. That said, SGD-optimized ensembling still requires $N$ forward passes through all experts, making it computationally expensive. This cost could be mitigated by imposing sparsity-inducing regularization in the SGD optimization. We leave this open for future research.

### 3.2.2 Distillation

As an alternative to full ensembling, we explore knowledge distillation (Tang et al., 2019; Clark et al., 2019; Khanuja et al., 2021), where a single model is trained to approximate the predictions of an ensemble. This approach retains the benefits of ensembling while significantly reducing inference cost.

We distill the best-performing ensemble from Section 3.2.1 into a single model by fine-tuning the pretrained Phi-2 model using LoRA to match the predictions of an ensemble of fine-tuned experts. This allows us to compress the ensemble into a single, lightweight model while preserving much of its performance. As shown in Figure 2, this distillation balances performance and computational efficiency, achieving slightly better accuracy than uniform averaging while requiring only a single forward pass.

While distillation reduces inference costs, it still requires two separate training stages: first optimizing the ensembling coefficients, then distilling the ensemble into a single model. This effectively doubles the total SGD optimization cost compared to direct model fusion.

We also explore a gradient-free alternative based on linear programming (LP) in Appendix B.6. The LP formulation optimizes the ensembling coefficients to minimize the worst-case multi-task error, and naturally produces sparse solutions where only 30% of the $\lambda_i$ are nonzero. While this sparsity is attractive for reducing inference cost, the LP solution achieves a higher average multi-task loss ($1.150 \pm 0.003$) compared to uniform ensembling ($0.990 \pm 0.002$), illustrating a trade-off between computational efficiency and average-case performance.

> **Key Takeaways**
>
> Uniform ensembling is a surprisingly competitive approach that can be further improved by directly optimizing the ensembling coefficients with SGD. Distillation reduces inference cost but adds significant training overhead. Oracle remains the best approach, leaving room for improvement.

## 4 Merging and Routing

Ensembling provides a strong baseline, but optimizing ensembling coefficients with SGD and distillation has diminishing returns. To further close the gap with the oracle baseline without a significant increase in the training and inference overhead, we explore merging and routing, which fuse model weights in parameter space.

Mode connectivity literature suggests that fine-tuned models often reside in a connected region of the loss landscape (Neyshabur et al., 2020; Verma & Elbayad, 2024), making merging and routing promising alternatives to ensembling. However, these claims are primarily focused on the single-task settings, leaving their applicability to multi-task learning uncertain. If effective, merging and routing could eliminate the need for multiple forward passes while retaining expert knowledge.

Merging combines LoRA parameters across experts, where $\lambda_i$ is *input-independent*. We examine both uniform merging, where all experts contribute equally, and learned merging, where $\lambda_i$ is optimized either globally or per layer via SGD. To motivate our experiments beyond uniform merging, we investigate mode connectivity in the multi-task setting. Routing extends merging by making $\lambda_i$ input-dependent. This provides additional adaptivity, potentially improving over both merging and ensembling without added inference cost.

### 4.1 Uniform Merging

A key consideration in merging is whether to average the LoRA factors $A$ and $B$ separately or to first reconstruct the full-rank update $W_i = A_i B_i$ and merge in full parameter space. Since LoRA introduces row and column permutations, merging directly in the low-rank subspace risks alignment issues between experts. However, because all experts originate from the same pretrained model and the same parameter initialization, we found that merging in the low-rank space yields comparable results to merging in full parameter space. As shown in Appendix B.3, both approaches perform similarly, leading us to adopt $(A, B)$ merging for computational efficiency. We note that while full-rank merging can sometimes slightly outperform low-rank merging (Appendix B.2), the gap between them is small relative to the merging-vs-routing gap that is our main focus, and low-rank merging avoids the costly reconstruction of full-rank updates.

Given $N$ experts, uniform merging is then performed as: $A^* = 1/N \sum_{i=1}^{N} A_i$ and $B^* = 1/N \sum_{i=1}^{N} B_i$. This method eliminates the need for multiple forward passes, reducing inference cost while preserving the shared structure across experts. However, our results in Figure 2 demonstrate that uniform merging significantly underperforms all ensembling methods and most baselines, with the exception of the shared expert baseline. This might suggest that the mode connectivity hypothesis may not hold in the multi-task setting, limiting the effectiveness of uniform merging. We further investigate this hypothesis in the next subsection.

### 4.2 Mode Connectivity

In the single-task setting, mode connectivity is typically defined as the absence of a loss barrier when interpolating between two models trained on the same task. Extending this concept to the multi-task setting is less straightforward. Given two experts fine-tuned on separate tasks, we consider two possible definitions of mode connectivity: (i) *global connectivity*, when there is no loss increase when interpolating between the two models on a dataset that mixes both tasks, and (ii) *task-specific connectivity*, when there is no loss increase for each task separately along the interpolation path.

The second definition is more restrictive, as a model trained on a given task is naturally expected to perform better on that task than an interpolated model. To investigate mode connectivity in our setting, we analyze

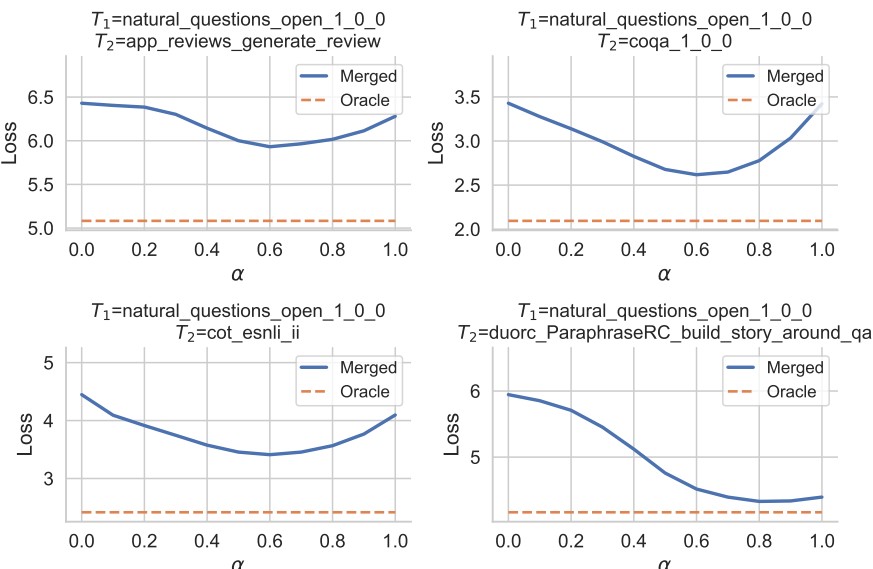

Figure 3: **Mode connectivity analysis in the multi-task setting.** For each subplot, we interpolate between two experts $(A_1, B_1)$ and $(A_2, B_2)$, independently fine-tuned on separate tasks $T_1$ and $T_2$ from the Flan v2 dataset (Longpre et al., 2023). We evaluate the performance of the interpolated model ($A_\alpha = (1-\alpha)A_1 + \alpha A_2, B_\alpha = (1-\alpha)B_1 + \alpha B_2$) on the combined datasets that contains both tasks $T_1 + T_2$, with $\alpha \in [0,1]$ shown on the x-axis. Let $\mathcal{M}_{O_1}$ and $\mathcal{M}_{O_2}$ represent the oracle experts that are best for tasks $T_1$ and $T_2$, respectively. Then the solid pink line represents the average performance of $\mathcal{M}_{O_1}$ and $\mathcal{M}_{O_2}$ on the combined dataset, where we use the best expert for each input. Our results demonstrate that careful expert selection outperforms linearly merging experts on multiple pairs of tasks.

pairwise interpolation paths between experts fine-tuned on different tasks. Given two experts $(A_1, B_1)$ and $(A_2, B_2)$ trained on separate tasks, we define the interpolated model as: $A_\alpha = (1-\alpha)A_1 + \alpha A_2$ and $B_\alpha = (1-\alpha)B_1 + \alpha B_2$ where $\alpha \in [0,1]$. We evaluate $(A_\alpha, B_\alpha)$ on the combined dataset from both tasks, comparing its performance to an oracle that selects the best expert for each input.

Figure 3 shows that while models appear linearly connected according to the *global connectivity* definition, selecting the right expert for each input leads to better performance. This suggests that merging alone may not be sufficient and motivates our investigation of learned merging coefficients and routing. We extend this analysis to 76 task pairs in Appendix B.1, finding that intra-cluster pairs have a mean loss barrier of 33.4% while cross-cluster pairs have a significantly larger barrier of 83.1%, which quantitatively motivates the effectiveness of MBC clustering.

### 4.3 Direct Optimization of Merging via SGD

An important design choice in merging is whether to use the same coefficients across all layers, referred to as global merging, or to allow them to vary per layer, referred to as layer-dependent merging. We learn both variants by optimizing $\lambda$ with SGD to minimize the multi-task loss:

$$\min_\lambda \frac{1}{|\mathcal{D}|} \sum_{(x_{<t}, x_t^*) \in \mathcal{D}} \ell\left(p(x_t \mid x_{<t}; A^*, B^*), x_t^*\right),$$

where $A^{*(l)} = \sum_{i=1}^N \lambda_i^{(l)} A_i^{(l)}$ and $B^{*(l)} = \sum_{i=1}^N \lambda_i^{(l)} B_i^{(l)}$ for each layer $l$.

Figure 2 shows that global merging unexpectedly outperforms layer-dependent merging, suggesting that enforcing consistency across layers may outweigh the benefits of additional flexibility. While SGD-optimized merging improves over uniform merging, it still falls short of ensembling, as shown in Figure 2. Since routing

can approximate the oracle in the extreme case where a single expert is selected, we next investigate whether it can close the gap with ensembling.

### 4.4 Is Routing Necessary?

To evaluate whether routing provides a significant advantage over merging and ensembling, we extend the optimization framework from the previous section by making $\lambda$ input-dependent in addition to layer-dependent. Specifically, we optimize the following objective:

$$\min_{\lambda} \frac{1}{|\mathcal{D}|} \sum_{(x_{<t}, x_t^*) \in \mathcal{D}} \ell\left(p(x_t \mid x_{<t}; A^*(x_{<t}), B^*(x_{<t})), x_t^*\right),$$

where $A^*(x, l) = \sum_{i=1}^{N} \lambda_i(x) A_i^{(l)}$ and $B^*(x, l) = \sum_{i=1}^{N} \lambda_i(x) B_i^{(l)}$ for input $x$ and layer $l$. Unlike merging, where $\lambda$ is fixed across inputs, routing allows $\lambda(x)$ to vary per input, providing greater flexibility.

As shown in Figure 2, our SGD-optimized routing not only outperforms all merging strategies but also surpasses all ensembling approaches and baselines except for the oracle. The remaining gap between routing and the oracle is likely due to the task-agnostic setting in which we operate.

**Routing architecture.** The routing module uses a per-layer linear gating architecture. For each expert $i$ and each routed projection group (Wqkv and out_proj), the router stores a learned signature vector $s_i^{(\ell)} \in \mathbb{R}^{d_{\mathrm{model}}}$. At each layer $\ell$, the routing coefficients for token $x$ are computed as $\lambda_i(x, \ell) = \mathrm{softmax}(s_i^{(\ell)\top} h_\ell(x))$, where $h_\ell(x)$ is the hidden state at layer $\ell$. Routing is performed at the token level: each token receives its own routing weights. Unlike Arrow, which derives its signatures from the SVD of the LoRA outer products, our SGD routing learns these signatures end-to-end via backpropagation. With $N = 10$ MBC experts, the total number of learned parameters is $N \times d_{\mathrm{model}} \times 2 \times L = 10 \times 2560 \times 2 \times 32 \approx 1.6$ million, which is comparable to a single LoRA expert ($\sim$2 million at rank 4). Scaling to the full $N = 256$ private library would increase this to $\sim$42 million parameters. See Table 1 for a complete comparison of computational costs.

Compared to Arrow, the strongest task-agnostic routing baseline, our SGD-optimized routing does not require selecting a subset of top-$k$ experts for routing, where only the $k$ experts with the highest routing logits contribute to the final prediction. While top-$k$ selection significantly impacts Arrow's performance, it has little effect on SGD-optimized routing (see Appendix B.5).

These results provide a clear answer to whether routing is necessary. The additional flexibility of routing significantly improves performance, nearly closing the gap with the oracle and highlighting the impact of learning expert contributions at the input level.

> **Key Takeaways**
>
> Both uniform and SGD-optimized merging underperform ensembling, suggesting a lack of task-specific mode connectivity. Our SGD-optimized routing approach nearly closes this gap, outperforming all merging and ensembling methods and falling just short of the oracle baseline.

## 5 From Private to Cluster Experts

In the previous sections, we showed that SGD-optimized routing achieves the best performance in the task-agnostic setting. However, learning input- and layer-dependent routing parameters introduces a significant computational cost. As shown in Table 1, routing over the 10 MBC experts requires $\sim$1.6 million router parameters and adds less than 1% inference overhead. However, scaling to the full library of $N = 256$ private experts would require $\sim$42 million router parameters (roughly 21$\times$ the size of a single LoRA expert). Specifically, for each LoRA layer, routing requires a learned transformation that maps the input to an expert combination distribution of the same dimensionality as the experts.

To mitigate this computational cost, we have so far used the 10 MBC experts (Ostapenko et al., 2024) instead of the full set of 256 private experts. It is important to distinguish between different forms of expert reduction studied in this paper. MBC experts are obtained by retraining one expert per cluster using aggregated task data, which requires access to both the training data and task identifiers. In contrast, the greedy subset selection analyzed in Figure 9 operates over the existing private experts without any retraining. Similarly, hierarchical clustering (HC) merges existing private experts within clusters via learned input-independent coefficients, also without retraining new experts. In this section, we re-evaluate whether reducing the number of experts through clustering is a viable alternative to using private experts. We also compare routing over MBC experts to a hierarchical clustering approach that groups tasks without requiring a separately trained expert per cluster. This analysis provides insight into whether expert selection strategies can retain strong multi-task performance while reducing computational complexity.

### 5.1 The Case for Expert Refactoring

To evaluate whether expert refactoring is a reasonable approach, we analyze the expert-by-task performance on a separate validation set. Our findings reveal that 58 out of 256 experts (about 30%) do not rank first on the very task they were fine-tuned on, suggesting that these experts may not be essential for optimal multi-task performance.

To further assess the impact of expert reduction on the average multi-task loss, we conduct a controlled experiment where we incrementally increase the number of experts and evaluate the best possible performance under task-level routing. Specifically, we greedily select the best expert for each of the 256 tasks given a limited set of $k$ experts and plot the average multi-task loss as a function of $k$. The results, shown in Figure 9, reveal that using only 150 of 256 experts (about 60%) already recovers the full average validation loss obtained under private oracle routing. These results motivate the use of compressed expert sets such as MBC.

### 5.2 Model Fusion with Private vs. MBC Experts

We compare private experts with MBC experts across oracle, uniform ensembling, uniform merging, and Arrow to assess the impact of expert reduction on model fusion. MBC experts (Ostapenko et al., 2024) cluster tasks based on LoRA parameter similarity, then train a single expert per cluster to reduce redundancy while preserving performance.

Figure 6 shows that MBC experts outperform private experts for oracle, uniform ensembling, and uniform merging, suggesting that expert clustering improves multi-task learning. However, top-4 Arrow performs better with private experts, likely because the top-4 expert selection inherently refactors experts, making explicit reduction less necessary.

### 5.3 Routing through Hierarchical Clustering

While MBC experts improve both efficiency and model fusion, they require retraining an expert per cluster using aggregated task data, which demands access to both the data and task IDs. As an alternative, we propose a hierarchical clustering routing approach that restructures the expert library without additional expert retraining.

Following the same Model-Based Clustering (MBC) approach, we first group experts by computing pairwise cosine similarities between their LoRA parameters. Instead of training a new expert for each cluster, we learn to merge the existing experts within each cluster using SGD in an input-independent manner. At the same time, we jointly learn a routing strategy to select between the merged cluster-level experts, optimizing expert selection at inference time while avoiding the need for retraining.

Figure 2 shows that while routing over hierarchical clustering (HC) does not match the performance of routing with MBC experts, which emphasizes the importance of expert retraining, initializing the routing matrices with Arrow's routing matrices (Arrow HC) improves the performance of hierarchical routing, surpassing all merging approaches.

> **Key Takeaways**
>
> MBC experts enhance efficiency and performance but require retraining per cluster. Hierarchical clustering offers a practical alternative by merging experts without retraining, albeit with some performance trade-off. Routing over MBC experts remains the best performing task-agnostic approach.

## 6 Related Work

**Loss Landscape and Mode Connectivity.** Understanding the structure of loss landscapes is central to model merging, particularly through the mode connectivity hypothesis, which suggests that models initialized identically can often be interpolated along low-loss paths (Freeman & Bruna, 2016; Garipov et al., 2018; Draxler et al., 2018; Frankle et al., 2020; Neyshabur et al., 2020; Wortsman et al., 2021; Benton et al., 2021; Juneja et al., 2022). Even when models trained on the same dataset do not naturally reside in the same basin, Entezari et al. (2021) hypothesize and empirically demonstrate that accounting for permutation invariance in neural networks reveals that solutions found by stochastic gradient descent (SGD) are likely linearly connected without loss barriers. Several subsequent works focus on identifying such permutations and transformations to align models within the same loss basin, enabling more effective merging (Singh & Jaggi, 2020; Ainsworth et al., 2022; Jordan et al., 2022; Peña et al., 2023).

While mode connectivity has enabled successful model merging in single-task settings, its applicability to multi-task learning remains understudied. Prior work on multi-task vision models suggests that mode connectivity may break down when models are fine-tuned on disjoint tasks (Yamada et al., 2023). Stoica et al. (2023) address this by introducing a feature-wise merging approach to fuse models trained on different tasks. We extend this line of work by examining mode connectivity in language models fine-tuned on different tasks.

**Model Ensembling.** Ensembling, which combines the outputs of multiple models to improve performance and robustness, has been widely used in machine learning to achieve state-of-the-art performance. Classical methods include stacking (Wolpert, 1992), bagging (Breiman, 1996), and boosting (Schapire et al., 1999), which have been successfully applied across various domains. In deep learning, Deep Ensembles (Lakshminarayanan et al., 2017) have become a cornerstone for uncertainty estimation and performance improvement by aggregating predictions from independently trained networks.

While ensembling is effective, its computational demands at inference time grow linearly with the number of models, making it challenging for resource-constrained settings such as large language models (LLMs). In this work, we evaluate the trade-off between the improved performance gained from ensembling and the related inference costs in the multi-task setting.

**Merging and Routing.** Merging aims to combine the parameters of individual models $w_1, \ldots, w_N$ into a single model $w^* = \sum_{i=1}^{N} \lambda_i w_i$ such that $\sum_{i=1}^{N} \lambda = 1$ and $\lambda_i \geq 0$, where $\lambda_i$ does not dependent on the input. Early merging methods, such as simple parameter averaging (McMahan et al., 2017; Stich, 2018), provide a baseline for merging models with similar architectures and initializations. More sophisticated merging approaches include Fisher Merging Matena & Raffel (2022), which weights parameters using the Fisher Information Matrix to prioritize components relevant to the task, and RegMean (Jin et al., 2022), which aligns model activations between the merged model and individual models to minimize output discrepancies.

Recent works, such as Matching Models in Task Subspaces (MaTS) (Tam et al., 2023), treat merging as a linear optimization problem to improve the performance across multi-task and intermediate-task settings. Routing extends merging by allowing the fusion coefficients $\lambda_i(x)$ to be input-dependent, introducing significant flexibility. Recent routing strategies range from static to learned approaches (Muqeeth et al., 2024; Yadav et al., 2024c; Tam et al., 2024b; Yadav et al., 2024b; Tam et al., 2024a), allowing for efficient single-task and multi-task learning. We highlight Arrow (Ostapenko et al., 2024) in particular, which introduces a task-agnostic routing method for LoRA experts, demonstrating that input-aware fusion can significantly enhance multi-task performance. For an extensive survey of ensembling, merging, and routing strategies, see Yadav et al. (2024a); Lu et al. (2024); Tam et al. (2024c).

## 7  Conclusion

In this work, we set out to empirically answer key questions about model fusion for multi-task learning, addressing them through a comprehensive evaluation, including:

*Do we observe gains when going beyond uniform ensembling or merging?* Our analysis indicates that while uniform ensembling is a simple and competitive approach, further optimizing the fusion coefficients yields additional gains. For merging however, the picture is different; we see that the different merging methods underperform ensembling. In summary, for settings where practitioners can afford a higher inference cost, ensembling is a reliable solution.

*When should practitioners use routing?* We have shown that for more cost-friendly inference, routing is a viable solution, outperforming both merging and ensembling while incurring a reasonable cost increase. When optimized, routing achieves the performance closest to the oracle.

*Quantitative trade-offs.* Our cost analysis (Table 1) reveals that the trade-offs are more nuanced than the qualitative picture suggests. With 10 MBC experts, SGD routing adds less than 1% inference compute overhead with only ∼1.6 million router parameters. Scaling to the full 256-expert library would increase the router to ∼42 million parameters (∼2% overhead), motivating expert reduction. Merging is free at inference but limited by mode connectivity constraints. Ensembling is reliable but requires $N$ forward passes ($10\times$ overhead with MBC experts).

*Limitations and future work.* Our experiments are conducted using a single base model (Phi-2) and dataset collection (Flan v2). While this setup is representative of the standard practical scenario for LoRA expert libraries, and the relative ordering of methods aligns with theoretical expectations about mode connectivity, validating our findings on additional base models, architectures, and task distributions is an important direction for future work.

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

# A   Experimental Details

We conduct our experiments using a library of LoRA experts fine-tuned from the Phi-2 model (Javaheripi et al., 2023) on the Flan v2 dataset (Longpre et al., 2023), made publicly available by Ostapenko et al. (2024). Each LoRA expert in the library was fine-tuned with a LoRA rank of 4, a dropout probability of 0.05, a LoRA scaling factor $\alpha = 16$, and a learning rate of $1 \times 10^{-4}$ with a linear warm-up followed by cosine annealing.

In addition to the full set of 256 task-specific experts, we also evaluate the Model-Based Clustering (MBC) experts from Ostapenko et al. (2024). These experts were obtained by applying MBC clustering to the private library, grouping tasks based on LoRA parameter similarity, then retraining a single expert per cluster. This results in a set of 10 MBC experts, each trained on a cluster of related tasks. Table 2 and Table 3 list the task assignments for each of the 10 clusters.

For all SGD optimization experiments, we tune the learning rate over $\{10^{-4}, 10^{-5}, 10^{-3}\}$ and train for 5 epochs. When using LoRA, we follow the same experimental setup as Ostapenko et al. (2024). During evaluation, we report the negative log-likelihood loss and compute the standard error across all 256 tasks.

## A.1   Derivation of Computational Costs

We derive the computational costs reported in Table 1 for the Phi-2 model with $d_{\mathrm{model}} = 2560$, $L = 32$ layers, LoRA rank $r = 4$, and $N = 10$ MBC experts (the setting used in our main experiments). We also provide the corresponding numbers for routing over the full set of $N = 256$ private experts. We count only linear-projection FLOPs, omitting attention-core compute (which is sequence-length dependent), LayerNorm, activations, and residuals, since these costs are constant across all methods we compare and do not affect the relative overheads reported.

**Base model cost per token per layer.** Each Phi-2 transformer layer contains an MLP block with two projections of shape $[10240 \times 2560]$ and $[2560 \times 10240]$, contributing $2 \times 10240 \times 2560 = 52.4$ million FLOPs per token. The attention projections consist of a fused QKV projection of shape $[2560 \times 7680]$ and an output projection of shape $[2560 \times 2560]$, contributing $2560 \times 7680 + 2560 \times 2560 = 19.7 + 6.6 = 26.2$ million FLOPs. The total base cost per token per layer is therefore $C_{\mathrm{base}} = 52.4 + 26.2 = 78.6$ million FLOPs.

**Routing overhead for SGD routing ($N$=10, soft routing).** The router computes a dot product between the hidden state ($d_{\mathrm{model}} = 2560$ dimensions) and every expert's signature vector ($N = 10$ experts) for each of the 2 routed projection groups (Wqkv and out_proj). This costs $N \times d_{\mathrm{model}} \times 2 = 10 \times 2560 \times 2 = 51,200$ FLOPs per token per layer. For the LoRA adapters, with all 10 experts active (soft routing), the KQV adapter costs $d_{\mathrm{model}} \times r + r \times d_{\mathrm{model}} \times 3 = 41,000$ FLOPs per expert, and the output adapter costs $d_{\mathrm{model}} \times r + r \times d_{\mathrm{model}} = 20,500$ FLOPs per expert. Across 10 active experts, the adapter cost is $10 \times (41,000 + 20,500) = 614,400$ FLOPs. The total routing overhead is $51,200 + 614,400 = 665,600$ FLOPs, or $665,600/78,600,000 \approx 0.85\%$ of $C_{\mathrm{base}}$.

**Router learned parameters.** The router stores one signature vector of size $d_{\mathrm{model}}$ per expert, per projection group, per layer: $N \times d_{\mathrm{model}} \times 2 \times L = 10 \times 2560 \times 2 \times 32 \approx 1.6$ million parameters. Each LoRA expert at rank $r = 4$ has $(d_{\mathrm{model}} \times r + r \times 3 \times d_{\mathrm{model}}) + (d_{\mathrm{model}} \times r + r \times d_{\mathrm{model}}) = 61,440$ parameters per layer, or $61,440 \times 32 \approx 2.0$ million parameters across all layers.

**Scaling to $N$=256 private experts.** If routing were applied directly over the full private expert library ($N = 256$, top-$k = 4$), the routing score cost would increase to $256 \times 2560 \times 2 = 1.31$ million FLOPs and the adapter cost to $4 \times 61,440 = 246,000$ FLOPs, yielding a total overhead of $\sim 1.56$ million FLOPs ($\sim 2.0\%$ of $C_{\mathrm{base}}$). The router would require $\sim 42$ million learned parameters, roughly $21\times$ a single expert. This motivates the expert reduction strategies studied in Section 5.

**HC and Arrow HC overhead.** Both HC and Arrow HC cluster the 256 private experts into 10 groups, learn intra-cluster merging coefficients, and jointly learn inter-cluster routing. At inference, they route between 10 cluster-level merged experts using the same routing architecture as SGD routing, so the inference overhead is identical: $\sim 0.85\%$ of $C_{\mathrm{base}}$. The extra learned parameters include the $\sim 1.6$ million routing parameters plus the intra-cluster merging coefficients (a small number of scalars). The difference between

HC and Arrow HC is the initialization: HC uses random initialization, while Arrow HC initializes the routing matrix from Arrow's SVD-derived prototypes.

## B   Additional Results

### B.1   Extended Mode Connectivity Analysis

In this section, we extend the mode connectivity analysis from Figure 3 to a larger and more diverse set of task pairs. Using interpolation data across 76 complete task pairs, we distinguish between *intra-cluster* pairs (both tasks belong to the same MBC cluster) and *cross-cluster* pairs (tasks from different clusters). Across all 76 pairs, the mean loss barrier is 56.9% (std 44.1%). Intra-cluster pairs exhibit a mean barrier of 33.4% (std 25.0%, $n = 40$), while cross-cluster pairs have a substantially larger mean barrier of 83.1% (std 45.9%, $n = 36$). These results quantitatively confirm that tasks within the same cluster are better connected in the loss landscape, which directly motivates MBC clustering as a strategy for expert reduction. Figure 4 and Figure 5 show representative examples of intra-cluster and cross-cluster interpolation paths, respectively.

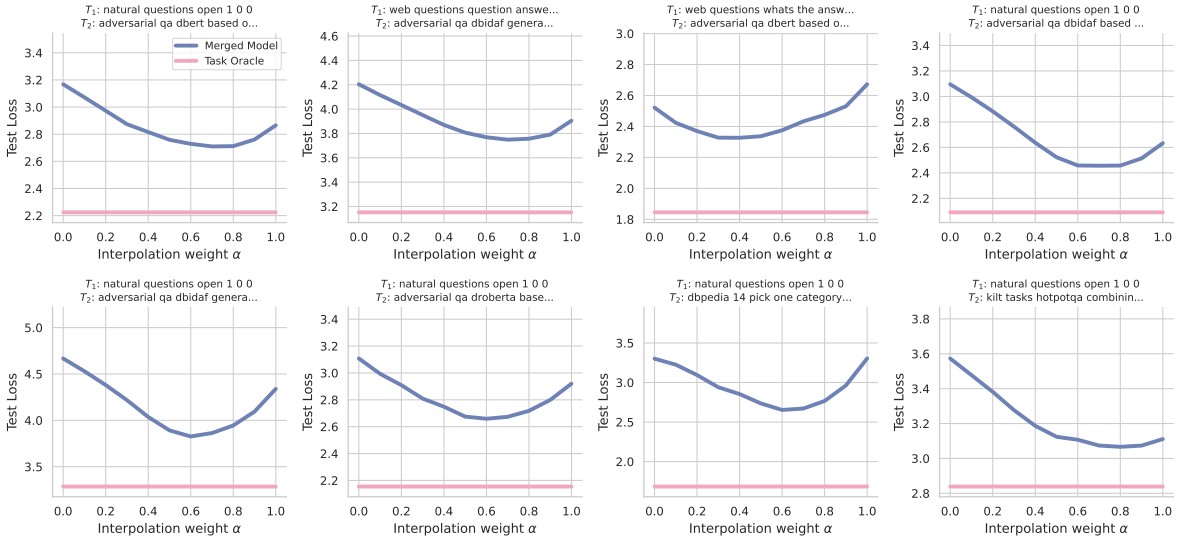

Figure 4: **Intra-cluster mode connectivity.** Interpolation paths between pairs of experts fine-tuned on tasks within the same MBC cluster. For each subplot, we interpolate between two experts $(A_1, B_1)$ and $(A_2, B_2)$ and evaluate the combined test loss on both tasks $T_1 + T_2$ as a function of the interpolation weight $\alpha$. The solid line shows the merged model loss, while the dashed line shows the oracle performance (selecting the best expert for each input). Intra-cluster pairs exhibit relatively smaller loss barriers (mean 33.4%, std 25.0%) compared to cross-cluster pairs (Figure 5), confirming that MBC clustering groups tasks that are better connected in parameter space.

### B.2   Private vs. MBC Experts Fusion

We compare model fusion performance when using the full set of private experts versus the MBC experts, which are obtained by clustering tasks and training a single expert per cluster. While the main text highlights key takeaways, here we provide a deeper analysis of why MBC experts generally perform better in ensembling and merging while underperforming in Arrow-based routing.

One critical difference between these settings is that MBC experts are explicitly trained to generalize across multiple tasks, whereas private experts are optimized for individual tasks. This additional generalization may explain why ensembling and merging benefit from MBC experts, as they introduce less variability in task-specific performance. Conversely, Arrow routing, which selects a sparse subset of experts per input,

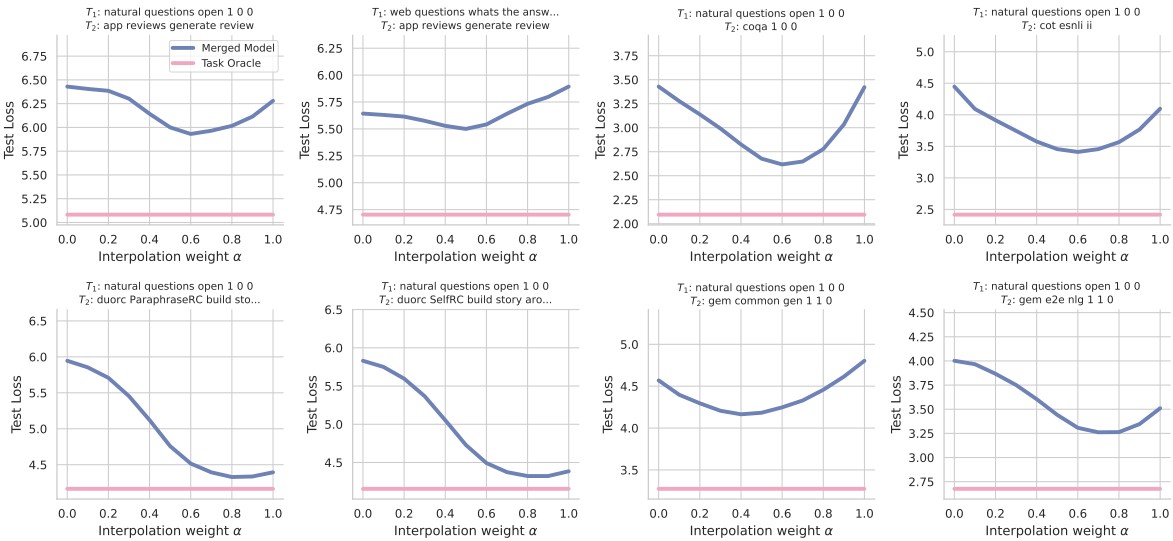

Figure 5: **Cross-cluster mode connectivity.** Interpolation paths between pairs of experts fine-tuned on tasks from different MBC clusters. The experimental setup is identical to Figure 4. Cross-cluster pairs exhibit substantially larger loss barriers (mean 83.1%, std 45.9%) than intra-cluster pairs, indicating that merging experts across cluster boundaries incurs a larger performance penalty. This gap motivates routing over merging for cross-cluster expert combinations and supports the use of MBC clustering to group tasks with compatible loss landscapes.

relies on the presence of highly specialized experts to maximize performance. Since MBC reduces the expert pool, it inherently limits the fine-grained selection capability that Arrow leverages.

To formalize Top-k Arrow, let $\lambda_{\mathrm{Arrow}}(x) \in \mathbb{R}^N$ represent the routing scores assigned to the $N$ experts for a given input $x$. Top-k Arrow restricts the routing to only the top $k$ experts with the highest scores, setting all other probabilities to zero and renormalizing within the selected subset:

$$
p_i(x) = \begin{cases} \dfrac{\exp(\lambda_{\mathrm{Arrow},i}(x))}{\sum_{j \in S_k(x)} \exp(\lambda_{\mathrm{Arrow},j}(x))}, & i \in S_k(x) \\ 0, & \text{otherwise} \end{cases}
$$

where $S_k(x)$ is the set of the top-$k$ experts for input $x$.

Figure 6 presents the performance comparison across different model fusion approaches, showing that while MBC experts enhance ensembling and merging, private experts remain more effective for Arrow routing, given the flexibility they provide for sparse expert selection.

## B.3 Full vs. Low-rank Merging

We compare two approaches to merging LoRA experts: merging in the full parameter space versus merging directly in the low-rank space. Given that each expert is fine-tuned using LoRA, we can either reconstruct the full-rank weight update as $W_i = A_i B_i$ before merging or merge the LoRA matrices $A_i$ and $B_i$ directly.

Formally, these two approaches are defined as follows:

- Full-rank merging: Compute $W_i = A_i B_i$ for each expert, then merge the full-rank updates via

$$
W^* = \sum_{i=1}^{N} \lambda_i W_i.
$$

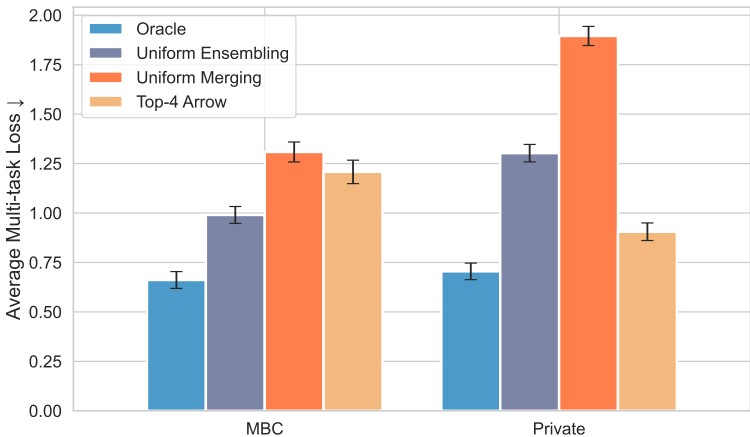

Figure 6: Performance comparison between private and MBC experts across different model fusion approaches.

- Low-rank merging: Merge the LoRA factors directly before reconstructing the update:

$$A^* = \sum_{i=1}^{N} \lambda_i A_i, \quad B^* = \sum_{i=1}^{N} \lambda_i B_i.$$

Since LoRA factorization introduces row and column permutations, merging in the low-rank space requires careful alignment of $A_i$ and $B_i$. However, because all experts originate from the same pretrained model, we find that low-rank merging performs comparably to full-rank merging.

We report results in Figure 7, showing that both methods yield comparable performance. For efficiency, we adopt low-rank merging in all of our experiments in this paper, as it avoids the costly reconstruction of full-rank updates.

### B.4 Ensembling Probabilities vs. Logits

We evaluate whether averaging model outputs at the probability level or the logit level leads to better performance in ensembling. Prior work suggests that averaging probabilities is preferable due to improved calibration, as logits can be uncalibrated and vary in scale across models. However, this claim has not been thoroughly validated in the multi-task expert ensembling setting, motivating our empirical analysis.

To compare both approaches, we compute the final ensemble prediction using:

- Probability-level ensembling: Averaging the softmax probabilities of each expert before selecting the most likely token.

- Logit-level ensembling: Averaging the pre-softmax logits and applying softmax afterward.

Figure 7 shows that probability-level ensembling indeed outperforms logit-level ensembling.

### B.5 Calibration of Arrow vs. SGD-Optimized Routing

We analyze the impact of calibration on Arrow routing compared to our SGD-optimized routing. While both methods aim to dynamically combine experts based on input-dependent routing coefficients, we observe significant differences in their sensitivity to the number of selected experts.

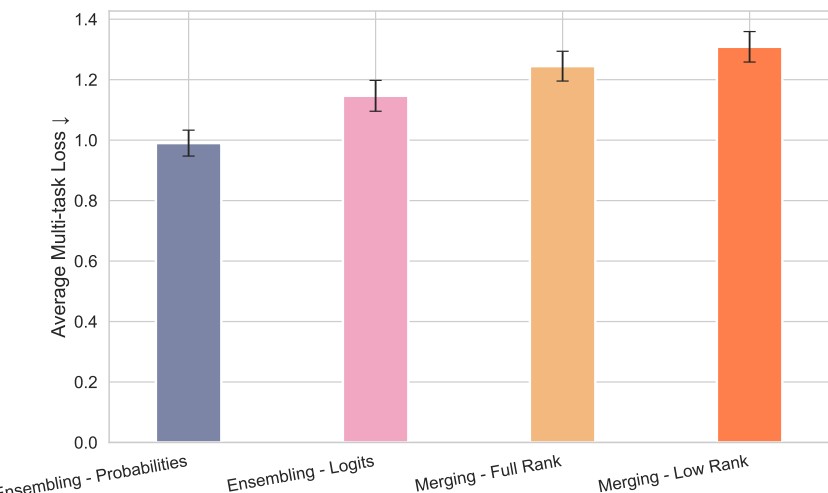

Figure 7: Comparison of multi-task test loss across different ensembling and merging strategies. Ensembling probabilities outperforms ensembling logits, confirming that averaging probabilities leads to better calibration. Similarly, full-rank merging slightly outperforms low-rank merging. Error bars indicate standard error over tasks.

One key metric for evaluating the robustness of a routing method is the performance gap between using a single expert (top-1) vs. using multiple experts (top-k). A large gap suggests that the routing method struggles to assign appropriate weights when limited to fewer experts, indicating poor calibration. Conversely, a small gap implies that the routing method consistently selects experts that yield stable performance.

Formally, top-k routing introduces a constraint where only the top $k$ experts with the highest routing coefficients contribute to the fused parameters. To enforce this sparsity, we define a *binary mask $m_i(x)$* such that:

$$m_i(x) = \begin{cases} 1, & \text{if } i \in \text{Top-k}(\lambda_1(x), \ldots, \lambda_N(x)) \\ 0, & \text{otherwise} \end{cases}$$

We then re-normalize the routing coefficients only among the top-k selected experts to preserve the sum-to-one constraint:

$$\lambda_i'(x) = \frac{m_i(x)\lambda_i(x)}{\sum_{j=1}^{N} m_j(x)\lambda_j(x)}.$$

The final routed model parameters are given by $A^*(x,l) = \sum_{i=1}^{N} \lambda_i'(x)A_i^{(l)}$ and $B^*(x,l) = \sum_{i=1}^{N} \lambda_i'(x)B_i^{(l)}$. This sparsification forces the model to rely on only the most relevant experts per input.

Our experiments show that the multi-task test loss delta between top-1 and top-4 routing is **0.303** for Arrow with MBC experts, whereas it is only **0.013** for SGD-optimized routing. This suggests that Arrow relies heavily on selecting the right combination of multiple experts to achieve strong performance, whereas SGD-optimized routing assigns more stable and well-calibrated expert weights, reducing its dependence on selecting a specific number of experts.

This analysis further highlights why SGD-optimized routing outperforms Arrow in our main results, as it is more robust to hyperparameter choices such as the number of selected experts, making it a more practical and reliable approach for multi-task model fusion.

### B.6 Approximate optimization via Linear Programming

In addition to the gradient-based approaches that we explore in Section 3.2 to learn ensembling coefficient, we explore a gradient-free alternative in this section. Namely, we formulate ensembling at the output level as a linear program (LP), optimizing the ensembling coefficients $\lambda$ to minimize the worst-case multi-task error. Given $N$ experts and $T$ tasks, let $M \in \mathbb{R}^{N \times T}$ be the validation error matrix computed using a heldout validation set, where $M_{it}$ is the error of expert $i$ on task $t$.

Our objective is to determine the optimal ensembling mixture that minimizes the worst-case performance across tasks,

$$\min_\lambda \max_q \lambda^T M q = \sum_{i=1}^N \lambda_i \sum_{t=1}^T q_t M_{it}.$$

Equivalently, the optimal ensemble coefficients $\lambda$ can be obtained by solving the following linear program (LP):

$$\min_{c,\lambda} \quad c \quad \text{s.t.} \quad \lambda_i \geq 0, \sum_{i=1}^N \lambda_i = 1, \sum_{i=1}^N \lambda_i M_{it} \leq c, \forall t.$$

We show in Figure 8 that the solution to this linear program results in only 30% of the coefficients $\lambda_i$ being nonzero. This sparsity suggests that the LP naturally selects a small subset of experts, effectively pruning redundant contributors. This property has important implications for reducing computational overhead, as it allows for expert selection without requiring explicit pruning strategies. However, this solution achieves an average multi-task loss of $1.150 \pm 0.003$, which is worse than uniform ensembling achieving a loss of $0.990 \pm 0.002$. This result suggests that simply solving for a worst-case optimal mixture does not necessarily yield the best average-case performance across tasks.

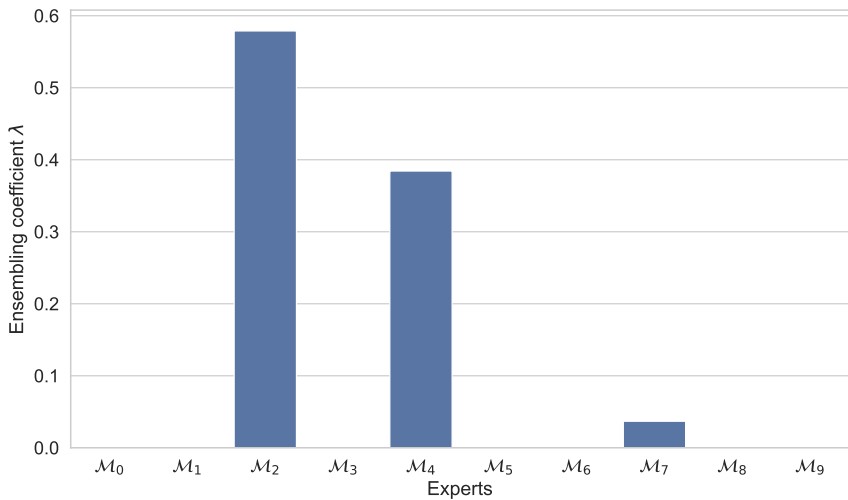

Figure 8: **Sparsity of Learned Ensembling Coefficients.** The LP solution leads to a sparse set of coefficients, with only 30% of the $\lambda_i$ being nonzero. This figure illustrates the distribution of learned coefficients across experts.

### B.7 Expert Selection

Given the computational cost of routing over a large pool of experts, it is important to determine whether a smaller, carefully selected subset can achieve comparable results.

To explore this, we conduct a greedy expert selection process, progressively adding experts based on their contribution to overall validation performance. Specifically, we start with the single best expert for each

task and iteratively select additional experts that yield the largest reduction in multi-task loss. At each step, we evaluate performance by selecting the best expert or best expert mixture for each task while still maintaining a task-agnostic setting.

Figure 9 illustrates the results of this selection process, showing how performance scales as we increase the number of experts. The curve demonstrates clear diminishing returns, with an elbow point where adding more experts yields little additional benefit. Notably, using only a fraction of the full expert pool is sufficient to recover near-optimal performance, suggesting that expert refactoring can significantly improve efficiency without sacrificing accuracy.

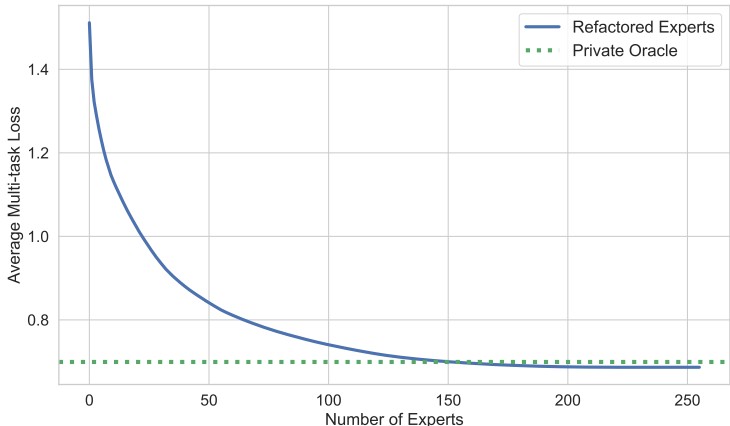

Figure 9: Performance as a function of the number of experts used in routing. The figure will illustrate diminishing returns and the elbow point where reducing experts retains near-optimal performance.

Table 2: Task assignments for clusters 1–5 obtained through Model-Based Clustering (MBC).

| Cluster | Assigned Tasks |
|---|---|
| Cluster 1 | "ropes_background_new_situation_answer", "ropes_prompt_bottom_no_hint", "ropes_plain_background_situation", "ropes_new_situation_background_answer", "ropes_given_background_situation", "ropes_prompt_beginning", "ropes_read_background_situation", "ropes_plain_bottom_hint", "ropes_plain_no_background", "ropes_prompt_mix", "ropes_prompt_bottom_hint_beginning", "ropes_background_situation_middle" |
| Cluster 2 | "glue_sst2_2_0_0", "adversarial_qa_droberta_generate_question", "true_case", "stream_qed", "huggingface_xsum", "race_middle_Write_a_multi_choice_question_for_the_following_article", "cot_esnli", "cot_gsm8k", "trec_1_0_0", "yelp_polarity_reviews_0_2_0", "lambada_1_0_0", "glue_cola_2_0_0", "ag_news_subset_1_0_0", "gem_dart_1_1_0", "math_dataset_algebra__linear_1d_1_0_0", "cnn_dailymail_3_4_0", "wiki_hop_original_explain_relation", "dbpedia_14_given_list_what_category_does_the_paragraph_belong_to", "gem_wiki_lingua_english_en_1_1_0", "fix_punct", "imdb_reviews_plain_text_1_0_0", "gigaword_1_2_0", "dbpedia_14_given_a_list_of_category_what_does_the_title_belong_to", "gem_web_nlg_en_1_1_0", "word_segment", "race_high_Write_a_multi_choice_question_for_the_following_article", "wmt16_translate_de_en_1_0_0", "cot_ecqa", "aeslc_1_0_0", "dream_generate_first_utterance", "wmt16_translate_fi_en_1_0_0", "dream_answer_to_dialogue", "para_crawl_enes", "adversarial_qa_dbert_generate_question", "race_middle_Write_a_multi_choice_question_options_given_", "wmt14_translate_fr_en_1_0_0" |
| Cluster 3 | "adversarial_qa_dbidaf_question_context_answer", "super_glue_record_1_0_2", "wiki_hop_original_generate_object", "adversarial_qa_droberta_tell_what_it_is", "dbpedia_14_given_a_choice_of_categories_", "wiki_hop_original_choose_best_object_affirmative_3", "wiki_hop_original_choose_best_object_interrogative_1", "adversarial_qa_dbert_answer_the_following_q", "wiki_hop_original_choose_best_object_affirmative_1", "wiki_hop_original_choose_best_object_interrogative_2", "adversarial_qa_droberta_question_context_answer", "squad_v2_0_3_0_0", "wiki_hop_original_generate_subject", "wiki_bio_guess_person", "adversarial_qa_dbidaf_answer_the_following_q", "adversarial_qa_droberta_answer_the_following_q", "adversarial_qa_dbert_tell_what_it_is", "race_high_Write_a_multi_choice_question_options_given_", "wiki_hop_original_choose_best_object_affirmative_2", "wiki_hop_original_generate_subject_and_object", "drop_2_0_0", "adversarial_qa_dbert_question_context_answer", "quac_1_0_0", "adversarial_qa_dbidaf_tell_what_it_is" |
| Cluster 4 | "wiqa_what_might_be_the_first_step_of_the_process", "wiqa_what_is_the_final_step_of_the_following_process", "wmt16_translate_ro_en_1_0_0", "wiqa_what_might_be_the_last_step_of_the_process", "wiki_bio_key_content", "gem_common_gen_1_1_0", "duorc_SelfRC_build_story_around_qa", "app_reviews_generate_review", "wiki_bio_what_content", "wiki_bio_who", "gem_e2e_nlg_1_1_0", "cot_esnli_ii", "wmt16_translate_tr_en_1_0_0", "wiqa_what_is_the_missing_first_step", "wiki_bio_comprehension", "coqa_1_0_0", "duorc_ParaphraseRC_build_story_around_qa", "multi_news_1_0_0" |
| Cluster 5 | "wiki_qa_found_on_google", "app_reviews_categorize_rating_using_review", "race_middle_Is_this_the_right_answer", "super_glue_cb_1_0_2", "wiki_qa_Topic_Prediction_Answer_Only", "wiki_qa_Direct_Answer_to_Question", "super_glue_wsc_fixed_1_0_2", "cot_gsm8k_ii", "unified_qa_science_inst", "race_high_Is_this_the_right_answer", "cot_strategyqa", "cot_ecqa_ii", "quarel_do_not_use", "wiki_qa_exercise", "wiki_qa_automatic_system", "cot_creak_ii", "quarel_heres_a_story", "quarel_choose_between", "stream_qed_ii", "wiki_qa_Topic_Prediction_Question_Only", "glue_qnli_2_0_0", "cot_sensemaking_ii", "super_glue_copa_1_0_2", "social_i_qa_Generate_the_question_from_the_answer", "social_i_qa_Show_choices_and_generate_index", "quarel_testing_students", "wiki_qa_Topic_Prediction_Question_and_Answer_Pair", "wiki_qa_Decide_good_answer", "wiki_qa_Jeopardy_style", "wiki_qa_Generate_Question_from_Topic", "definite_pronoun_resolution_1_1_0", "wiqa_effect_with_label_answer", "glue_wnli_2_0_0", "cot_qasc", "cot_strategyqa_ii", "quarel_logic_test", "stream_aqua_ii" |

Table 3: Task assignments for clusters 6–10 obtained through Model-Based Clustering (MBC) (continued).

| Cluster | Assigned Tasks |
|---|---|
| Cluster 6 | "quoref_Context_Contains_Answer", "duorc_SelfRC_generate_question_by_answer", "quoref_Find_Answer", "duorc_ParaphraseRC_movie_director", "duorc_ParaphraseRC_answer_question", "quoref_Found_Context_Online", "quoref_Read_And_Extract_", "duorc_ParaphraseRC_title_generation", "duorc_ParaphraseRC_decide_worth_it", "quoref_What_Is_The_Answer", "duorc_ParaphraseRC_generate_question", "quoref_Guess_Title_For_Context", "quoref_Answer_Test", "duorc_SelfRC_question_answering", "duorc_SelfRC_title_generation", "duorc_ParaphraseRC_generate_question_by_answer", "duorc_ParaphraseRC_extract_answer", "duorc_SelfRC_answer_question", "duorc_SelfRC_decide_worth_it", "duorc_ParaphraseRC_question_answering", "quoref_Answer_Question_Given_Context", "duorc_SelfRC_extract_answer", "quoref_Guess_Answer", "quoref_Answer_Friend_Question", "duorc_SelfRC_movie_director", "duorc_SelfRC_generate_question", "quoref_Given_Context_Answer_Question" |
| Cluster 7 | "super_glue_rte_1_0_2", "cot_sensemaking", "super_glue_wic_1_0_2", "cos_e_v1_11_rationale", "cos_e_v1_11_generate_explanation_given_text", "anli_r3_0_1_0", "dream_generate_last_utterance", "paws_wiki_1_1_0", "cot_creak", "stream_aqua", "snli_1_1_0", "cos_e_v1_11_i_think", "glue_qqp_2_0_0", "cos_e_v1_11_explain_why_human", "anli_r2_0_1_0", "anli_r1_0_1_0", "glue_stsb_2_0_0", "cos_e_v1_11_aligned_with_common_sense", "glue_mnli_2_0_0", "social_i_qa_I_was_wondering", "cosmos_qa_1_0_0", "glue_mrpc_2_0_0", "social_i_qa_Generate_answer" |
| Cluster 8 | "dream_read_the_following_conversation_and_answer_the_question", "app_reviews_convert_to_star_rating", "cos_e_v1_11_question_option_description_text", "social_i_qa_Show_choices_and_generate_answer", "quartz_answer_question_based_on", "sciq_Direct_Question_Closed_Book_", "qasc_qa_with_separated_facts_3", "quartz_given_the_fact_answer_the_q", "quartz_answer_question_below", "kilt_tasks_hotpotqa_final_exam", "sciq_Multiple_Choice", "wiqa_does_the_supposed_perturbation_have_an_effect", "wiki_qa_Is_This_True_", "quartz_use_info_from_question_paragraph", "cos_e_v1_11_question_description_option_text", "sciq_Direct_Question", "cos_e_v1_11_question_option_description_id", "qasc_qa_with_separated_facts_2", "app_reviews_convert_to_rating", "wiqa_effect_with_string_answer", "qasc_qa_with_separated_facts_5", "wiqa_which_of_the_following_is_the_supposed_perturbation", "quartz_having_read_above_passage", "cos_e_v1_11_question_description_option_id", "qasc_qa_with_separated_facts_1", "cos_e_v1_11_description_question_option_text", "qasc_qa_with_combined_facts_1", "qasc_is_correct_1", "cos_e_v1_11_description_question_option_id", "social_i_qa_Check_if_a_random_answer_is_valid_or_not", "sciq_Multiple_Choice_Closed_Book_", "quartz_use_info_from_paragraph_question", "qasc_is_correct_2", "qasc_qa_with_separated_facts_4", "quartz_read_passage_below_choose", "quartz_paragraph_question_plain_concat", "dream_baseline", "sciq_Multiple_Choice_Question_First" |
| Cluster 9 | "race_middle_Read_the_article_and_answer_the_question_no_option_", "race_high_Select_the_best_answer", "quail_description_context_question_answer_id", "quail_context_question_description_text", "race_high_Select_the_best_answer_no_instructions_", "quail_context_description_question_answer_id", "race_high_Read_the_article_and_answer_the_question_no_option_", "race_high_Taking_a_test", "super_glue_multirc_1_0_2", "race_middle_Select_the_best_answer", "quail_context_question_description_answer_id", "quail_description_context_question_answer_text", "quail_context_question_answer_description_text", "race_high_Select_the_best_answer_generate_span_", "race_middle_Select_the_best_answer_generate_span_", "quail_context_question_answer_description_id", "quail_context_description_question_answer_text", "quail_context_description_question_text", "quail_context_question_description_answer_text", "quail_description_context_question_text", "race_middle_Taking_a_test", "quail_no_prompt_id", "quail_no_prompt_text", "race_middle_Select_the_best_answer_no_instructions_" |
| Cluster 10 | "natural_questions_open_1_0_0", "web_questions_whats_the_answer", "web_questions_question_answer", "dbpedia_14_pick_one_category_for_the_following_text", "kilt_tasks_hotpotqa_combining_facts", "web_questions_short_general_knowledge_q", "kilt_tasks_hotpotqa_straighforward_qa", "adversarial_qa_dbidaf_generate_question", "adversarial_qa_droberta_based_on", "web_questions_get_the_answer", "kilt_tasks_hotpotqa_complex_question", "web_questions_potential_correct_answer", "trivia_qa_rc_1_1_0", "kilt_tasks_hotpotqa_formulate", "adversarial_qa_dbert_based_on", "adversarial_qa_dbidaf_based_on", "squad_v1_1_3_0_0" |

