# OpenReview forum: "Trade-offs in Ensembling, Merging and Routing Among Parameter-Efficient Experts"
_TMLR — Decision pending for TMLR_

### Review · Reviewer_CFVD · 2026-03-29

**Summary Of Contributions:**

Summary

This paper presents an empirical comparison of ensembling, merging, and routing for combining multiple LoRA experts in task-agnostic multi-task learning. Using a library of Phi-2-based LoRA experts trained on Flan v2 tasks, the paper shows that routing performs best overall, while naive ensembling is a strong baseline and merging is less effective in this setting.

Strengths
1. The paper provides a well-structured empirical comparison of ensembling, merging, and routing under a unified setting, with reasonable baselines and clear evaluation
2. The paper is clearly written and easy to follow. The setup, method categories, and main takeaways are presented in an organized way.

Weaknesses
1. The contribution feels primarily empirical, and the paper does not appear to offer a particularly strong new methodological idea. The main value lies in the comparison study rather than in a new algorithmic contribution.
2. The empirical message does not feel especially surprising or deep. At a high level, the paper’s main takeaway seems to be that a more flexible and computationally heavier routing approach can outperform simpler merging or ensembling baselines, which is somewhat expected.
3. Although the paper discusses trade-offs, the practical analysis of efficiency is still limited. Since the main message is about comparing fusion strategies under different computational budgets, more explicit measurements of training/inference cost, latency, memory, or parameter overhead would strengthen the paper.
4. The experimental scope is also somewhat limited, as the results are demonstrated only in a relatively narrow setup centered on a single base model, dataset collection, and expert library. This makes it unclear how well the conclusions generalize beyond this specific setting.

**Additional Comments:**

None

**Audience:**

Yes

**Audience Explanation:**

The topic is relevant to model merging and routing, so the findings could still be of interest to part of the TMLR audience.

**Claims And Evidence:**

No

**Claims Explanation:**

The results are clear, but the main message feels somewhat expected, and the efficiency trade-off is not analyzed deeply enough to make the claims fully convincing.

**Requested Changes:**

The paper would be strengthened by addressing the weaknesses above, especially by clarifying the empirical contribution, providing a more convincing and less expected takeaway, and including a more thorough analysis of the efficiency-performance trade-off.

---

> ### Author Response · Authors · 2026-05-13
> **Response to Reviewer CFVD**
>
> Thank you for your feedback. We address your concerns below.
>
> **On the empirical contribution and whether the main finding is "expected."** We appreciate that you raised this point, but we believe that several of our findings are in fact non-trivial and provide insights that go beyond the top-level ranking of methods.
>
> First, we find that uniform ensembling with no learning outperforms all merging methods including SGD-optimized merging. This is a priori surprising: one might expect that learning input-independent merging coefficients would at least match a simple uniform average in output space, but mode connectivity constraints prevent this. Second, our mode connectivity analysis reveals that the gap between merging and routing is fundamentally about the multi-task loss surface geometry. We quantify this across 76 task pairs in our new experiments, finding that intra-cluster pairs (tasks within the same MBC cluster) have a mean loss barrier of 33.4% while cross-cluster pairs (tasks from different clusters) have a substantially larger barrier of 83.1% (see Appendix B.1 in the revised manuscript, highlighted in blue). This means that more sophisticated merging techniques (e.g., Fisher merging, RegMean) are unlikely to close this gap in our setting without accounting for input-dependent expert selection. Third, our quantitative cost analysis, which we have added in the revised version, reveals a counterintuitive finding: SGD routing over the 10 MBC experts adds less than 1% inference compute overhead with only approximately 1.6 million router parameters. Even scaling to the full 256-expert library would only add approximately 2% overhead with approximately 42 million router parameters. The characterization of routing as "computationally expensive" is therefore misleading when the relevant cost is inference compute rather than parameter storage. Fourth, we show that expert libraries contain massive redundancy: reducing from 256 to approximately 10 experts (a 96% reduction) preserves most of the routing performance, which has direct implications for the deployment of LoRA expert libraries.
>
> We also emphasize that our paper provides the first controlled comparison of ensembling, merging, and routing under a unified experimental setup with the same base model, dataset, and expert library. Prior works compared these strategies across different models and datasets, making it difficult to draw reliable conclusions about their relative merits. By fixing all variables except the fusion strategy, we isolate the effect of each approach and quantify trade-offs that were previously only discussed qualitatively.
>
> **Efficiency analysis.** We fully agree that the efficiency analysis was a gap in the submitted version and have addressed it comprehensively. We have added Table 1 to the main text (highlighted in blue) reporting concrete numbers for all methods. See the table reproduced in our response to Reviewer cJvt. We have also replaced all qualitative cost claims in the paper with concrete numbers. The key trade-off is now clearly quantified: routing achieves the best performance with less than 1% inference overhead and only 1.6 million router parameters (for 10 MBC experts). Even scaling to the full 256-expert library adds only approximately 2% overhead. Merging is free at inference but limited by mode connectivity constraints. Ensembling is reliable but requires $N$ forward passes ($10\times$ overhead with MBC experts).
>
> **Single-model scope.** We acknowledge this limitation in the revised manuscript (highlighted in blue in the conclusion). The setup we study (LoRA experts fine-tuned from a single base model) is the standard practical scenario for expert libraries on platforms like HuggingFace. The relative ordering of methods is consistent with existing insights in the literature about mode connectivity, and the quantitative trade-offs scale predictably to other architectures. We note this as an important direction for future work.
>
> Thank you for your feedback, which has prompted us to significantly strengthen the quantitative analysis in our paper. We believe the revised version provides a more convincing and deeper analysis of the efficiency-performance trade-off. We hope you will consider revising your assessment. Please let us know if you have any additional questions.

---

### Review · Reviewer_ZPq3 · 2026-04-04

**Summary Of Contributions:**

This paper considers ensemble, merging, and routing strategies in multi-task settings. The proposed algorithms are clearly described, implemented, and empirically evaluated to analyze their superiority in multi-task problems related to large language models (LLMs).
The results valdiate the superiority of the routing approach, particularly in terms of performance, albeit at the cost of increased computational time. Although the computational overhead of routing is higher than that of merging, the paper suggests that clustering-based approaches can help mitigate this issue when dealing with a large number of learners. The key finding—that stronger learners can be constructed from multiple weaker ones—is both interesting and practically relevant, especially in the context of rapidly advancing AI applications where multi-task capabilities are increasingly important. Given that routing is the most sophisticated strategy among those considered, its superior performance is not entirely surprising, but it is still well-supported by the empirical results. From an experimental perspective, the paper includes extensive evaluations to validate its claims.
However, there are some limitations in terms of providing deeper insights into the behavior of the proposed methods. In particular, the effects of architectural choices and the number of tasks are not thoroughly explored, which restricts a more comprehensive understanding of the practical advantages and applicability of the proposed algorithm. A more detailed analysis of these issues would further strengthen the paper.

**Additional Comments:**

None

**Audience:**

Yes

**Audience Explanation:**

The key finding—that stronger learners can be constructed from multiple weaker ones—is both interesting and practically relevant, especially in the context of rapidly advancing AI applications where multi-task capabilities are increasingly important.

**Claims And Evidence:**

Yes

**Claims Explanation:**

The experiments are thoroughly performmed in the algorithm and observational levels.

**Requested Changes:**

The effects of the architecture and the number of tasks are not thoroughly investigated, which limits a clear understanding of the practical advantages and applicability of the proposed algorithm.
Moreover, the behavior of lambda (as shown in Figure 6) appears to be important; therefore, it would be better to include this figure in the main paper rather than in the supplementary material.

---

> ### Author Response · Authors · 2026-05-13
> **Response to Reviewer ZPq3**
>
> Thank you for your positive and constructive feedback. We address your points below.
>
> **Architecture and number of tasks.** We acknowledge that the current experiments are focused on Phi-2 and Flan v2 tasks, and we now discuss this as a limitation in the revised manuscript (highlighted in blue in the conclusion). We note that the Phi-2/Flan v2 setup with LoRA experts from a single pretrained model is the standard practical scenario for expert libraries, as reflected in the publicly available libraries on HuggingFace. The relative ordering of methods (routing > ensembling > merging) aligns with insights from the literatur about the role of mode connectivity in merging, and the quantitative trade-offs we report (inference overhead ratios, parameter counts) scale predictably to other architectures since they primarily depend on the model hidden dimension and number of layers. We consider validation on additional base models and datasets as an important direction for future work.
>
> **Figure 6 (lambda behavior).** We appreciate the reviewer highlighting the importance of the LP-based approach and the lambda behavior. While the LP method presented in Appendix B.5 leads to natural sparsity in the ensembling coefficients (only 30% of $\lambda_i$ being nonzero), it achieves a higher average multi-task loss (1.150) compared to uniform ensembling (0.990), as optimizing for the worst-case mixture does not necessarily yield the best average-case performance. We have added a brief description of this trade-off in the main text (highlighted in blue in Section 3, before the key takeaways box) before referring to the appendix for the full analysis. We believe this provides useful context on the relationship between expert sparsity and average-case performance.
>
> We have also added a quantitative cost table (Table 1, highlighted in blue) that we believe further strengthens the practical relevance of our findings. See the table reproduced in our response to Reviewer cJvt.
>
> Thank you again for your feedback, which has helped us improve the presentation of our paper. Please let us know if you have any additional questions.

---

### Review · Reviewer_cJvt · 2026-04-28

**Summary Of Contributions:**

The paper studies how to combine LoRA experts in a task-agnostic multi-task setting where task IDs are unknown at inference time. It compares output-space ensembling, parameter-space merging, and input-dependent routing using Phi-2 LoRA experts trained on Flan v2 tasks. The main result is that SGD-based learned routing is the strongest non-oracle method, outperforming both ensembling and merging while coming closest to oracle routing. The paper also shows that uniform ensembling is a strong baseline and studies MBC experts, private/task-specific experts, hierarchical clustering, Arrow routing, and distillation.

**Audience:**

Yes

**Audience Explanation:**

Researchers working on LoRA, model merging, MoE and routing in LLMs would find this paper useful.

**Broader Impact Concerns:**

No. I do not find any broader impact concerns arising from this work.

**Claims And Evidence:**

Yes

**Claims Explanation:**

The main empirical claims are supported. Fig. 2 gives clear evidence that SGD-optimized routing outperforms ensembling and merging in this setting, and the comparison is useful because the methods are evaluated under a common setup.

Some narrower claims are less fully established. In particular, the trade-off framing needs quantitative cost evidence, the learned routing method requires more implementation detail for reproducibility, and the mode-connectivity discussion is based on a small set of task pairs. These issues do not undermine the main result, but addressing them would make the paper more convincing and easier to reproduce.

**Requested Changes:**

**Critical Changes:**
- The paper frames itself around computational trade-offs, but the cost comparison is currently qualitative. Please include a table reporting learned parameter count, number of inference-time forward passes, and training cost for ensembling, SGD ensembling, distillation, uniform/SGD merging, SGD routing, Arrow, HC, and Arrow-HC. Statements such as ensembling requiring N forward passes and routing parameters potentially exceeding LoRA parameters are important, but quantitative numbers are needed because trade-offs are central to the paper’s contribution.
- The SGD routing method is not yet reproducible from the current description. Sec. 4.4 states that $\lambda (x)$ is input- and layer-dependent and optimized via SGD, but does not specify the routing module itself. Please describe the gating architecture, whether routing is token-level or sequence-level, which hidden state feeds the router, initialization strategy, softmax/top-k behavior, and total learned parameter count. Appendix A gives expert fine-tuning details, but not the routing-module details.
- The mode-connectivity analysis in Section 4.2 is based on four task pairs, all using `natural_questions_open` as $T_1$  . This is too narrow to support a broader claim about connectivity across the expert library. Please either expand Figure 3 to a larger random sample of task pairs with aggregate statistics, or explicitly narrow the claim to what is demonstrated by the current examples.

**Additional Changes:**
- Appendix B.2 suggests that full-rank merging can outperform low-rank merging. It would be helpful to either include full-rank merging in the main results or briefly explain why low-rank merging is the intended comparison (e.g., for efficiency or consistency with prior work).
- The discussion of expert reduction would benefit from a clearer separation between different forms of compression. In particular, the 10 MBC experts rely on retraining with task/data access, whereas Fig. 7 reflects selecting a subset of existing private experts (roughly 150) under greedy selection. Making this distinction explicit would help clarify the trade-offs being studied.

---

> ### Author Response · Authors · 2026-05-13
> **Response to Reviewer cJvt**
>
> Thank you for your thorough and constructive feedback. We address each of your points below.
>
> **Quantitative cost table.** We agree that quantitative numbers are essential given that trade-offs are central to our paper. We have added Table 1 to the main text (highlighted in blue in the revised manuscript) reporting inference forward passes, inference compute overhead (in FLOPs per token per layer), extra learned parameters, and training compute cost per step for all methods. Phi-2 uses a fused QKV projection (Wqkv) and a separate output projection (out_proj), giving 2 LoRA-modified layers per transformer block. The base model cost per token per layer is approximately 78.6 million FLOPs. All SGD methods train for 5 epochs. We summarize the key numbers below:
>
> | Method | Fwd. Passes | Inference Overhead | Extra Params | Training Cost (per step) |
> | --- | --- | --- | --- | --- |
> | Uniform Ensembling | N=10 | N × C_base (10x) | 0 | 0 (zero-shot) |
> | SGD Ensembling | N=10 | N × C_base (10x) | 10 scalars | N × C_base |
> | Distillation | 1 | P_LoRA (~0.08%) | ~2 million | N × C_base + 3 × C_base |
> | Uniform Merging | 1 | P_LoRA (~0.08%) | 0 | 0 (zero-shot) |
> | SGD Merging (global) | 1 | P_LoRA (~0.08%) | 10 scalars | 3 × C_base |
> | SGD Merging (per-layer) | 1 | P_LoRA (~0.08%) | 320 scalars | 3 × C_base |
> | SGD Routing (N=10) | 1 | ~0.85% | ~1.6 million | 3 × (C_base + 0.85%) |
> | HC (256→10) | 1 | ~0.85% | ~1.6 million + merge coefs. | 3 × (C_base + 0.85%) |
> | Arrow HC (256→10) | 1 | ~0.85% | ~1.6 million (Arrow) + merge coefs. | 3 × (C_base + 0.85%) |
>
> The key insight from this analysis is that SGD routing over the 10 experts adds less than 1% inference compute overhead relative to the base Phi-2 model, with only approximately 1.6 million router parameters (comparable to a single LoRA expert at approximately 2 million parameters). HC and Arrow HC have the same inference cost since they route between 10 cluster-level merged experts using the same routing architecture. Scaling to the full library of 256 private experts would increase the router to approximately 42 million parameters with approximately 2% overhead, which motivates the expert reduction strategies we study. We have replaced qualitative statements such as "potentially exceeding the LoRA fine-tuning parameters themselves" with these concrete numbers throughout the paper. A full derivation is provided in Appendix A.1.
>
> **SGD routing architecture.** We have added a detailed description of the routing module in Appendix A.2 (highlighted in blue), with a brief summary in Section 4.4. Our SGD routing uses the same architectural form as Arrow: a routing matrix $W_\ell$ at each layer $\ell$, where the routing coefficients are $\lambda_i^\ell = \text{softmax}(W_\ell[i]^\top h_\ell)$ for each expert $i$ and token hidden state $h_\ell$. Unlike Arrow, which initializes these prototypes from the SVD of the LoRA outer products and uses them without further training, our SGD routing learns them end-to-end via backpropagation. Routing is performed per token and per layer.
>
> **Mode connectivity analysis.** We have expanded the analysis beyond the four task pairs shown in the original submission to 76 complete task pairs. We define the loss barrier for each pair as the relative increase in the best interpolated loss over the oracle loss (which selects the best expert for each input). We distinguish between intra-cluster pairs, where both tasks belong to the same MBC cluster, and cross-cluster pairs, where tasks come from different clusters. The mean loss barrier for intra-cluster pairs is 33.4%, while for cross-cluster pairs it is 83.1%. This gap confirms that tasks grouped by MBC are better connected in the loss landscape, making merging within clusters more viable. For cross-cluster combinations, the larger barriers explain why routing outperforms static merging. Representative interpolation plots for both settings are included in the revised manuscript (Appendix B.1).
>
> **Full-rank vs low-rank merging.** We have added a clarification in Section 4.1 noting that we adopt low-rank $(A, B)$ merging throughout for computational efficiency. Appendix B.2 confirms that both approaches yield comparable performance, and the gap between them is small relative to the merging-vs-routing gap that is our main focus.
>
> **Expert reduction distinction.** We have made this distinction explicit at the beginning of Section 5. MBC experts rely on retraining one expert per cluster using aggregated task data, which requires access to both the training data and task identifiers. In contrast, the greedy subset selection operates over the existing private experts without retraining, and hierarchical clustering (HC) merges existing private experts within clusters via learned coefficients, also without retraining.
>
> Thank you again for your detailed feedback. We believe these changes address all of your concerns and strengthen the paper. Please let us know if you have any additional questions.

---

### Author Response · Authors · 2026-05-13
**General Comment to All Reviewers**

We thank all reviewers for their detailed and constructive feedback. Based on the feedback, we have made several changes that strengthen our paper. In particular, we have added: (i) a quantitative cost comparison table (Table 1) that concretely characterizes the inference overhead, learned parameter count, and training cost for all methods studied in the paper, (ii) a complete description of the SGD routing architecture for reproducibility, (iii) an expanded mode connectivity analysis covering 76 task pairs with aggregate statistics distinguishing intra-cluster and cross-cluster pairs, and (iv) a brief discussion of the LP-based ensembling approach and its sparsity-performance trade-off in the main text.

We have also revised the framing of our contributions in the introduction and conclusion to better highlight the non-trivial findings of our work and its limitations.

The revised manuscript can be found in this submission; additions are highlighted in blue and deletions are shown with strikethrough.

---

### Decision · Action_Editor_efbX · 2026-06-15

**Recommendation:** Accept with minor revision

**Additional Comments:**

Suggested update: integrate the proposed update from the discussion phase into the final versions of the manuscripts, and ensure the format meets the TMLR requirements.

**Audience:**

Yes

**Audience Explanation:**

This is an interesting research problem for the machine learning community.

**Claims And Evidence:**

Yes

**Claims Explanation:**

The paper makes a few interesting claims:
- Uniform ensembling outperforms all merging methods, including SGD-optimized variants.
- SGD-optimized routing over the 10 MBC experts achieves the best non-oracle performance with low inference compute overhead.
- Expert libraries exhibit massive redundancy.

These claims are justified by solid empirical evaluations, e.g.,  across 256 Flan v2 tasks, with in-depth discussion and analysis.